# Striatal Patchwork of D1-like and D2-like Receptors Binding Densities in Rats with Genetic Audiogenic and Absence Epilepsies

**DOI:** 10.3390/diagnostics13040587

**Published:** 2023-02-05

**Authors:** Evgeniya T. Tsyba, Inna S. Midzyanovskaya, Lidia M. Birioukova, Leena M. Tuomisto, Gilles van Luijtelaar, Kenul R. Abbasova

**Affiliations:** 1Faculty of Biology, Lomonosov Moscow State University, 119991 Moscow, Russia; 2Institute of Higher Nervous Activity and Neurophysiology of RAS, 117485 Moscow, Russia; 3School of Pharmacy, University of Eastern Finland, 70211 Kuopio, Finland; 4Donders Centre for Cognition, Radboud University, 6500 HE Nijmegen, The Netherlands

**Keywords:** genetic generalized epilepsy, animal epilepsy models, mixed form of epilepsy, KM rats, WAG/Rij rats, audiogenic epilepsy, convulsive seizures, non-convulsive seizures, D1-like DA receptors, D2-like DA receptors

## Abstract

Binding densities to dopamine D1-like and D2-like receptors (D1DR and D2DR) were studied in brain regions of animals with genetic generalized audiogenic (AGS) and/or absence (AbS) epilepsy (KM, WAG/Rij-AGS, and WAG/Rij rats, respectively) as compared to non-epileptic Wistar (WS) rats. Convulsive epilepsy (AGS) exerted a major effect on the striatal subregional binding densities for D1DR and D2DR. An increased binding density to D1DR was found in the dorsal striatal subregions of AGS-prone rats. Similar changes were seen for D2DR in the central and dorsal striatal territories. Subregions of the nucleus accumbens demonstrated consistent subregional decreases in the binding densities of D1DR and D2DR in epileptic animals, irrespective of epilepsy types. This was seen for D1DR in the dorsal core, dorsal, and ventrolateral shell; and for D2DR in the dorsal, dorsolateral, and ventrolateral shell. An increased density of D2DR was found in the motor cortex of AGS-prone rats. An AGS-related increase in binding densities to D1DR and D2DR in the dorsal striatum and motor cortex, areas responsible for motor activity, possibly reflects the activation of brain anticonvulsive loops. General epilepsy-related decreases in binding densities to D1DR and D2DR in the accumbal subregions might contribute to behavioral comorbidities of epilepsy.

## 1. Introduction

Epilepsy is a chronic brain disorder characterized by an enduring predisposition to generate seizures. Epileptic seizures may develop due to an imbalance of excitatory and inhibitory neurotransmitters. Glutamate and gamma-aminobutyric acid are the main neurotransmitters playing a crucial role in the pathophysiology of this balance. Accumulating evidence indicates that monoamines are another group to regulate seizure activity. Experimental studies have indicated that the basal ganglia and its major neurotransmitter dopamine (DA) are involved in seizure propagation [1] and play a role as a “remote control system” over seizures’ duration [2] Importantly, epilepsy-related shifts in brain aminergic balance can be a substrate for a number of psychiatric and behavioral comorbidities [3,4] seen in epileptic patients.

Animal models are not replaceable for experimental approaches to studying brain pathologies [5]. One of the oldest animal models for generalized convulsive epilepsies is the Krushinsky-Molodkina (KM) rat, which is genetically prone to severe audiogenic seizures [6]. Rats of the KM strain demonstrate increased sound sensitivity, leading to generalized convulsive seizures in response to acoustic provocation. The main role in generating audiogenic seizures (AGS) belongs to the brain stem, including the corpora quadrigemina [7].

Absence epilepsy is a nonconvulsive, generalized form of epilepsy. Behavioral symptoms of absence seizures (AbS) include a recurrent decrease in the level of consciousness, interruption of ongoing activity, and in some cases, minor myoclonisms. WAG/Rij (Wistar Albino Glaxo rats from Rijswijk) rats are an inbred strain with genetic predisposition to absence epilepsy [8]. Spontaneous generalized and bilateral symmetrical spike-and-wave discharges (SWDs), with a frequency of 7–10 Hz, are recorded on the EEG in all adult WAG/Rij rats [9]. The model is used worldwide and has face, construct, and pharmacological validity [10]. Up to one third of adult WAG/Rij rats show convulsive seizures in response to an intense sound stimulation [11]. This particular subpopulation of WAG/Rij rats is a model of a mixed (convulsive and nonconvulsive) form of epilepsy [12] and shares principal features with “pure” absence and audiogenic epilepsy models. In clinical practice and in WAG/Rij rats, absence seizures can be aggravated by the intake of standard anticonvulsants such as phenytoin [13]. Pharmacotherapy for mixed forms of epilepsy is often complicated. Therefore, it is necessary to delineate common neuronal and neurochemical substrates of both types and mixed forms of epilepsy, ensuring epileptogenesis itself and its psychiatric comorbidities, frequently seen in patients, are understood.

According to the focal cortical theory, SWDs are primed in the peri-oral region of the somatosensory cortex and maintained by the reticular nucleus and relay nuclei of the thalamus [14,15]. It has been shown that the hyperexcitability of the somatosensory cortex of epileptic WAG/Rij rats is due to a decrease in GABA-mediated inhibition [9]. The dopaminergic (DA) system, not involved directly in SWD initiation, modulates spike-wave discharges’ threshold outside the cortico-thalamo-cortical loops. Experimental data indicate functional insufficiency of DA-ergic system in the genetic absence-epileptic rats [16,17]. A reduced cataleptic threshold was evident in WAG/Rij rats that were stressed or injected with small doses of haloperidol [16,18]. AGS-prone KM rats were also sensitive to haloperidol-induced catalepsy; this effect was more prominent in rats with the highest intensity of AGS [19,20]. Signs of DAergic system dysfunction were found in KM rats as well: the level of DA in the dorsal striatum was higher in KM rats than in Wistar ones, and the level of DA reached its maximum level much slower in KM rats after the injection of both the indirect DA-mimetic amphetamine and D2/D3 antagonist raclopride than in Wistar ones [21].

It is suggested that the DA system is a part of the brain’s anti-epileptic system. This hypothesis is based on studies showing that inhibition of nigrotectal projection in SNpr stops all seizure types in different genetic and pharmacological epilepsy models [22]. In the GAERS model, closely resembling the WAG/Rij model as a genetic absence model, administration of D2-like receptor selective agonists (quinpirole and bromocriptine) in the core of the nucleus accumbens (NAcbC) reduced and the antagonist (sulpiride, haloperidol) enhanced SWDs. The combination of D1-like (D1DR) and D2-like receptors (D2DR) selective agonists showed a stronger inhibitory effect compared to separate administration of the drugs [17]. Previous autoradiography studies comparing DA receptors distribution in WAG/Rij and ACI (epilepsy-resistible rats) brains showed significant higher binding density to D2DR in WAG/Rij at parietal and frontal cortex (motor and somatosensory areas), lower binding densities to D2DR in dorsal caudate-putamen (CPu) and hippocampus, as well as decreased binding densities to D1DR in the NAcbC and CPu [23].

In the present study, four groups will be compared (Figure 1). This will allow us to determine whether generalized epilepsies have a distinct DA receptor binding profile, or whether this is different in the mixed model, or due to being AGS sensitive, or having only generalized absence seizures.

## 2. Materials and Methods

### 2.1. Animals

The study included twenty male rats, 6–8 months old, weighing 290–410 g: WS (*n* = 5), WAG/Rij (*n* = 5), WAG/Rij-AGS (*n* = 5), KM (*n* = 5). Animals were housed in individual cages under natural light and dark conditions with free access to food and water. The rats were habituated to the experimental room three days before the audiogenic susceptibility test. The test lasted 90 s (‘key ringing’, 13–85 kHz, 50–60 dB). AGS were rated according to Krushinsky’s scale [11]. All experiments were carried out in accordance with Directive 2010/63/EU on the protection of animals used for scientific purposes [24]. Rats were decapitated under deep general anesthesia. Coronal sections (18 µm) were cut by cryotome at −18 °C at the following anatomical levels: +2.4; +1.68; −1.08; −2.16 regarding the bregma for D2DR, and +2.28; +1.8; −0.96; −1.92 for D1DR [25].

### 2.2. D1-like Receptor Autoradiography

Brain regional D1DR-binding densities were examined as described previously [23,26]. Slide-mounted sections were incubated in Tris-buffer (50 mM Tris–HCl containing 120 mM NaCl, 5 mM KCl, 2 mM CaCl_2_, and 1 mM MgCl_2_; pH 7.7) containing 0.2 nM [3H]SCH 23390 (specific activity 66.0 Ci/mmol, Amersham (Amersham, UK)) for 90 min at room temperature. Sections for non-specific binding were incubated in Tris-buffer containing 0.2 nM [3H]SCH 23390 and 10−7 M cis-flupenthixol. After incubation, the slides were drained, washed twice for 5 min in buffer at +4 °C and briefly dipped twice into distilled water (+4 °C). Sections were dried at room temperature overnight and exposed to a tritium-sensitive film (3H Hyperfilm^®^, Amersham) at −20 °C for 6 weeks, together with Amersham 3H Microscale Autoradiography Standards^®^.

### 2.3. D2-like Receptor Autoradiography

Brain regional D2DR-binding densities were examined as described previously [23,26]. Slide-mounted sections were incubated in Tris-buffer (50 mM Tris–HCl containing 120 mM NaCl; pH 7.4) containing 0.4nM [3H]spiperone (specific activity 109.0 Ci/mmol, Amersham) for 60 min at room temperature. Sections for non-specific binding were incubated in Tris-buffer containing 0.4 nM [3H]spiperone, 10−5 M haloperidol, and 10−5 M ketanserin. After incubation, the slides were drained, washed twice for 5 min in buffer at 4 °C and briefly dipped twice into distilled water (4 °C). Sections were dried at room temperature overnight and exposed to a tritium-sensitive film (3H Hyperfilm^®^, Amersham) at −20 °C for 3 weeks, together with Amersham 3H Microscale Autoradiography Standards^®^. After the development, all films were digitized, and the images were processed in ImageJ.

### 2.4. Measurements

The brain structures were identified according to Paxinos and Watson (2007), as described above (Section 2.1). Measurements were carried out in the dorsal and ventral striatum, cingulate (Cg), motor (M), and somatosensory (S) cortex. Somatosensory, motor, and cingulate areas of the neocortex were measured without division into the primary and secondary cortex. The dorsal and ventral striatum were divided into subregions (Figure 2). The CPu was virtually divided into nine subregions, as shown in Figure 2. NAcb was divided into shell and core parts (abbreviated below as NAcbC and NAcbSh), further parcellated as shown on Figure 2. The obtained values of optical densities were converted into pmol/g of tissue by using the microscale standards (see Section 2.1). The obtained values of optical densities were converted into pmol/g of tissue by using the microscale standards (see Section 2.1).

### 2.5. Statistical Analyses

Statistical analysis was performed with Statistica14.0.0.15″ (TIBCO Software Inc., Pao Alto, CA, USA). Values are shown as means ± SEM. Analysis of D1DR and D2DR binding densities in a defined brain region was performed for each anatomical level and region separately, using ANOVA GLM. Two measurements were taken within each site of each hemisphere and averaged. To account for sources of variation caused by local imperfections in the films, the local background levels were taken as continuous GLM predictors; their effects are not reported below. Anatomical levels were analyzed separately; anatomical subregions of a defined structure (i.e., parts of cortex, CPu, NAcbC, and NAcbSh) were taken as within-subjects factors, so a repeated measures analysis was used. ANOVA GLM for regional data was run with the epilepsy type (AGS and/or AbS, 2 × 2 design) as two between-subjects factors. This analysis provided information about the general effect of AGS/AbS susceptibilities, as previously done [24]. The putative effects of AGS were checked by comparing the pooled group of KM and WAG/Rij-AGS rats with the pooled group of AGS-unsusceptible rats (i.e., WS and WAG/Rij). The effects of AbS’ proneness were estimated by comparing the pooled groups of WAG/Rij and WAG/Rij-AGS rats with the pooled groups of WS and KM rats. The general effect of epilepsy (“Epi”) was assessed by comparing normal WS rats with the pooled data of the three groups of epileptic rats (i.e., KM, WAG/Rij, and WAG/Rij-AGS). General ANOVAs were followed by Fisher LSD post-hoc tests, if needed. The minimal level of significance was set at *p* = 0.05. Local striatal gradients in binding densities to D1DR and D2DR were assessed in normal Wistar rats by pairwise comparisons of dependent samples (Wilcoxon test; the minimal *p* level was set at 0.05). For the correlation part of the study [27], the striatal and accumbal sets of local binding densities to D1DR and D2DR were averaged, thus sets of the mean individual values of D1- and D2-like binding densities were generated for CPu and NAcb. Spearman’s rank order for the inter-correlations between individuals’ dorsal and ventral striatal values of binding densities was calculated for each group of rats separately. The minimal level of significance was set at *p* = 0.05.

## 3. Results and Discussion

### 3.1. Distribution of D1DR and D2DR Binding Densities in Normal WS Rats

Generally, the regional distribution of binding densities for D1DR and D2DR agrees with those reported earlier [23,28,29]. Representative autoradiographic images are shown on Figure 3b, together with the mean subregional values of D1DR and D2DR binding densities for normal WS rat brain (Figure 3a).

### 3.2. Effects of Epilepsies

Recently, we have shown that there might be both epilepsy-type-specific changes in brain aminergic receptors and general (unspecific to seizure type) effects of epilepsy [27]. Below, we also map both epilepsy type-specific and type-unspecific effects (see the scheme for comparisons on Figure 1) on D1DR and D2DR binding densities in brain subregions of rats with genetic generalized epilepsies. The epilepsy-related changes composed a complex “patchwork” pattern, where sites of seizure-type-unspecific changes were surrounded by type-specific local alterations of binding densities. This pattern is similar to that seen for H3 histamine receptor binding densities in subregions of the prefrontal cortex [27] and putatively points to diffuse compensatory post-convulsive changes in brain aminergic activity.

#### 3.2.1. Dorsal Striatum

##### Striatal Binding Densities to D1DR

The rat strain did not differ, and did not tend to differ in their local binding densities to D1DR measured within the CPu at the most rostral level (AP = +2.28). All four groups showed the same pattern of lateromedial and ventrodorsal gradients (Figure 4, left panels), with maximal values at the ventrolateral borders of the CPu. Namely, the strain effect was not significant, *p* > 0.10; within-subject repeated measures ANOVA GLM showed a tendency to subregions’ effect, *p* = 0.09) These gradients were studied in detail in Wistar samples (Figure 3). Pairwise comparisons showed that for D1DR, the most dorsomedial locus (number 3, Figure 2) expressed the minimal binding densities (red downward triangle on Figure 3a), being significantly lower than the lateral sites (all *p*’s < 0.05 for comparisons with loci 1 and 6–9, level I; all *p*’s < 0.05 for comparisons with loci 1–7, level II). The most ventrolateral subregion (site number 7, Figure 2) demonstrated the highest binding density to D1DR (green upward triangle on Figure 3a), being significantly higher than the central and median sites (all *p*’s < 0.05 with loci 1–7, level I; all *p*’s < 0.05 for comparisons with loci 1–9, level II).

This lateromedial and ventrodorsal gradient is clearly visible (see Figure 4) and should be taken into account in experimental approaches as a source of internal variation. The next anatomical level (II, aimed at AP = +1.8) displayed a sharp AGS-related increase in D1DR binding densities, clearly localized at the dorsal border of the CPu. Namely, subregions were different {F(8.112 = 2.6), *p* = 0.01}, and strain*subregion interaction was significant as well {F(24.112 = 1.6), *p* = 0.04}. Closer analysis showed that the two AGS-susceptible groups were responsible for this main effect (AGS effect, F(1.14) = 4.9, *p* = 0.04), with a significant interaction of AGS * subregion (F(8.112 = 2.4, *p* = 0.02)). Post-hoc Fisher LSD tests were applied to estimate the local differences between the AGS-prone and AGS-resistant cohorts (Figure 1). The localized measurements (Figure 2) let us point out the subregions 1–3 as significantly elevated in AGS-prone subgroups (all p’s < 0.05, deep blue for sites 1, 2, and 3 in Figure 2 and Figure 4). The same tendency was seen in the neighborhood as well (all *p*’s < 0.10, light blue for sites 4, 5, and 6 in Figure 2 and Figure 4). The absence epilepsy cohort (i.e., WAG/Rij and WAG/Rij-AGS pooled together, see Figure 1) did not differ significantly from its controls (KM and WS rats pooled). Previously, we found that WAG/Rij rats displayed decreased D1DR binding densities in CPu of WAG/Rij rats, as compared to epilepsy-resistant ACI rats [23]. Here, we did not see the same effect; therefore, we couldn’t confirm that absence-epilepsy-related processes evoke changes in D1DR of the dorsal striatum.

##### Striatal Binding Densities to D2DR

The rat strains did not differ significantly in their local binding densities to D2DR measured at the most rostral level (AP = +2.28). However, the subregions were highly heterogeneous (within-subject repeated measures, subregion effects, F(8.112) = 8.0 and 12.7 for levels I and II, respectively; both *p*’s < 0.001), with significant strain*subregion interaction effect (F(24.112) = 4.2 and 4.6, for levels I and II, respectively; both *p*’s < 0.001) These gradients were checked in Wistar rats (Figure 3). Pairwise comparisons showed that for D2DR, the most dorsomedial locus (number 3, Figure 2) expressed the minimal binding densities (red downward triangle on Figure 3a), being significantly lower than the lateral sites (all *p*’s < 0.05 for comparisons with loci 1–9, level I; all *p*’s < 0.05 for comparisons with loci 1–7, level II). The most ventrolateral subregion (site number 7, Figure 2) demonstrated the highest binding density to D2DR (green upward triangle on Figure 3a), being significantly higher than the central and median sites (all *p*’s < 0.05 with loci 2–3 and 5–9, level I; all *p*’s < 0.05 for comparisons with loci 1–3 and 5–9, level II). Closer analysis revealed that AGS-susceptible groups differed from their AGS-resistant controls (Figure 1 explains the comparison), with a significant AGS * subregion interaction effect (F(8.112) = 6.6, *p* < 0.001). Post-hoc Fisher LSD tests pointed to the two central loci (numbers 3 and 6 on Figure 2 and Figure 4) with an AGS-related increase in D2DR (both *p*’s = 0.01 for comparisons with the AGS-resistant cohort). The next anatomical level (II, AP = +1.8) demonstrated a similar pattern, with the dorsomedial AGS-related increase in D2DR (Figure 4, right panels). The local striatal measurements differed from each other: since subregions showed a significant effect (F(8.112) = 12.7, *p* < 0.001), with a significant strain * subregion interaction (F(24.112) = 4.6, *p* < 0.001) and a major effect of AGS * subregion interaction (F(8.112) = 7.8, *p* < 0.001). To be more precise, subregions 3, 6 demonstrated the AGS-related increase (*p*’s = 0.01 and 0.04, respectively; deep blue spots on Figure 4), neighbored by the same effect and the same tendency in subregions 5 and 2 (*p*’s = 0.04, 0.05, respectively; deep and light blue spots in Figure 4). For audiogenic epilepsy, however, it is known that the hierarchical neuronal network, which determines this pathology, does not involve the basal ganglia [30]. It allows us to speculate that DA receptor density elevation in the dorsal striatum is a consequence of seizures and a possible reason for comorbid catalepsy. A previous study reported a reduced D2 receptor density in the isolated homogenized striatum of KM rats [31]. Therefore, this contradiction could be attributed to the heterogeneity of the striatum (see above) and to the different seizure status in the experimental groups: the cited paper describes seizure-naïve KM rats. Our rats had three sound provocations that resulted in seizure fits in the AGS-susceptible groups (see Methods, Section 2.1), to mimic an active (but not yet kindled) seizure state. It has been shown that the D2 receptor density in the striatum decreases after acute seizures, while antiepileptic treatment increases the D2DR density [32]. Therefore, the reported AGS-related increase in D2DR should be attributed to a post-convulsive activation of the DA receptor system. One important conclusion from our study is about the natural heterogeneity of the striatum, both dorsal and ventral. This should be taken into account during the interpretation of literature data and the planning of further experiments. The comparison of absence epilepsy-prone and absence epilepsy-free groups (Figure 1) did not reveal any significant difference in binding densities to D2DR. Thus, we did not confirm our previous finding on a decreased D2DR binding density related to absence epilepsy and judged by the comparison of WAG/Rij and ACI rats [23].

#### 3.2.2. Ventral Striatum

##### Accumbal Binding Densities to D1DR

The rat strains did not differ significantly in their binding densities to D1DR at the two anatomical levels studied. The accumbal subregions were highly heterogeneous, (F(3.45) = 7.2, *p* < 0.001) and (F(5.80) = 34.4, *p* < 0.001) for levels I and II, respectively. No epilepsy-related changes were seen at level I (AP = +2.28). More caudally, at level II (AP = +1.8), the pooled epileptic cohort (i.e., joined KM, WAG/Rij, and WAG/Rij-AGS groups compared to WS groups; see Figure 1) tended to be lower than the control rats (*p* = 0.10). A post-hoc test revealed that ventrolateral and dorsal aspects of the shell (*p* = 0.02 and *p* < 0.001 in Fisher LSD tests), as well as the dorsal core (*p* = 0.03) displayed epilepsy-related decrease in D1DR (magenta spots on Figure 5, left panels). This was accompanied by the same tendency to decrease, seen in the AGS-prone groups (medial shell, medial core, light blue spots on Figure 5).

The results agree with the previous finding on a decreased D1DR binding density in the NAcbC of WAG/Rij rats [23]. As mentioned earlier, in the GAERS model, administrations of D1/D2 agonists into this particular structure were effective in modulating SWDs [17]. Therefore, we can hypothesize that the mediadorsal aspects of the NAcbC are engaged in the brain’s anti-epileptic loop, which is responsible for the occurrence of SWDs. Our study extends the effect to convulsive epilepsy as well, since AGS-prone rats demonstrated the same pattern of decreased binding to D1DR seen in accumbal regions (Figure 5 left panels).

##### Accumbal Binding Densities to D2-like Dopamine Receptors

The rat strains did not differ significantly in their binding densities to D2DR at the two anatomical levels studied. No epilepsy-related changes were seen at level I (AP = +2.28). More caudally, at level II (AP = +1.8), a complex pattern of epilepsy-related decreases was seen. Interactions between subregion * strains were significant {F(15.75) = 2.1, *p* = 0.02}. A seizure-type specific decrease was seen in a single structure: AGS-groups demonstrated a dramatic decrease in dorsolateral shell (*p* = 0.02; Figure 5 right panels), mostly at the expense of KM rats (deep blue column on a chart). A general effect of epilepsies was the decrease in binding densities to D2DR seen in the dorsal shell and ventrolateral shell (*p*= 0.04 and *p* = 0.01, respectively; magenta spots on the scheme, Figure 5), paralleled by the same tendency seen for the medial shell (*p* = 0.07).

The shell of NAcb was not a sensitive site to modulate SWDs in the GAERS model [17], and no difference in D2DR was seen earlier [23]. However, in epileptic patients, this particular region is reported as a putative site for deep brain stimulation in intractable partial epilepsy [33], and seizure-related neuronal degeneration was seen in the structure in patients with temporal lobe epilepsy [34]. Recently, it was shown that experimental temporal lobe seizures in mice induced by intra-amygdalar kainic acid administration increased the neuronal excitability of medium spine neurons of NAcbSh, both D1R-expressing and D2R ones [35]. Besides the fact that the pharmacology and the brain structures are completely different between temporal lobe epilepsy and absence epilepsy, it is also possible that the shell of the NAcb is not involved in ongoing seizure activity but is a constituent part of a slow anti-epileptic brain loop. Recently, behavioral characteristics were proposed as potential biomarkers for epileptogenesis in rats [36]. Therefore, DAergic transmission within NAcb is likely involved in shaping the behavioral comorbidities of epilepsies and thus might be an important target for pharmacological interventions.

An intriguing possibility would be to link the deficient D2DR binding densities (Figure 5 right panels), seen in the ventrolateral shell of the nucleus accumbens of epileptic rats, profoundly in KM rats, with recently reported social behavioral deficits in KM rats [37]. In human patients, the volume of the nucleus accumbens positively correlates with the size of the individual social network [38]. Therefore, accumbal deficits and degeneration, related to epilepsies, might contribute to the increased frequency of social deficits in epileptic patients [39]. Behavioral depression and anhedonia, described in WAG/Rij rats [40,41], generally attributable to absence epilepsy-related shifts in tissue indices of brain aminergic turnover [42], also might be related to the observed deficient D2DR binding densities within medial and dorsal aspects of accumbal shell (Figure 5, right panel). Further behavioral studies with local drug administration would shed more light on this important question.

#### 3.2.3. Motor and Somatosensory Cortex

Rat strains did not differ significantly in D1DR and D2DR binding densities measured at all four levels. No epilepsy related changes were seen in the D1DR binding densities in the cortical areas’ measures (joined primary and secondary motor cortices, somatosensory cortex).

In contrast, binding densities to D2-like DA receptors demonstrated a remarkable AGS-related elevation in the motor cortex (Figure 6; level III, AP = −0.96). The AGS effect was significant (F(1.15) = 5.8, *p* = 0.02), as was the AGS * subregion interaction (F(1.15) = 4.7, *p* = 0.04). Post-hoc tests showed that the pooled AGS cohort had a higher D2DR binding density in the motor cortex (*p* < 0.01). No difference was seen for absence-epileptic rats, in contrast to what was reported earlier [23]. Previously, ACI rats were used as the control group and it cannot be excluded that the choice of a non-WS derived control might have contributed to the earlier obtained differences. Another possibility is that perhaps a more detailed analysis targeted specifically to the barrel cortex of WAG/Rij rats would bring more information about possible alterations of DA receptors around “the hot spot”, the cortical site of origin of the SWDs. From the other side, it is also known that some WS rats have SWDs as well [43]. Therefore, the reason for the absence of a significant difference between WS and WAG/Rij rats could be the presence of SWDs at least in some WS rats or the choice of the non-epileptic control strain. For future experiments, it might be recommended to check not only the audiogenic seizures but also the presence of SWDs in all experimental groups.

### 3.3. Regional Correlations of D1DR and D2DR Binding Densities

Strong positive (Figure 7, all Rs >0.70) correlations between the mean local striatal and accumbal values of D1DR and D2DR binding densities (see Section 2.5) were seen in the control WS rats as well as in the WAG/Rij group, suggesting a highly functional coupling of local receptor systems. Putatively, a brainstem DA supply coordinates exposure and bindings to D1DR and D2DR within the projected regions, CPu and NAcb. Remarkably, this intimate relationship coupling was broken in the two AGS-prone groups (Figure 6) where mean accumbal binding densities to D2DR receptors lost its strong and positive correlation with their D1DR areas in KM and WAG/Rij-AGS rats (Figure 7). Although correlations [44] provide only suggestive information on linkage between the correlated units, we might hypothesize that having AGS experience imbalances receptor response to the brainstem DA supply specifically within the nucleus accumbens. The difference between the dopaminergic reactivity of the nigrostriatal and mesolimbic systems is essential for the formation of the complex behavioral phenotype and should further contribute to epileptogenesis as well [45].

## 4. Conclusions

WAG/Rij and KM strains are both WS-derived [6,43]. It allows us to suggest that regional differences in D1DR and D2DR binding densities could be attributed to the difference between convulsive and non-convulsive forms of epilepsy or its specific comorbidities, since our strains differ in the presence or absence of particular forms of epilepsy.

Our results point to the essential heterogeneity of studied striatal territories in terms of local binding densities to D1DR and D2DR. This should be taken into account in the interpretation of data in the literature.

Decreased D1DR binding density in the dorsal aspect of the nucleus accumbens core is a consistent feature of epileptic brains reported in literature and confirmed in our study. The obtained results suggest that this effect is not specific for absence-epileptic brains, but expands also on convulsive epilepsy. This particular structure might be a part of the brains’ anti-epileptic loop which in turn mediates behavioral comorbidities of epilepsies.

## Figures and Tables

**Figure 1 diagnostics-13-00587-f001:**
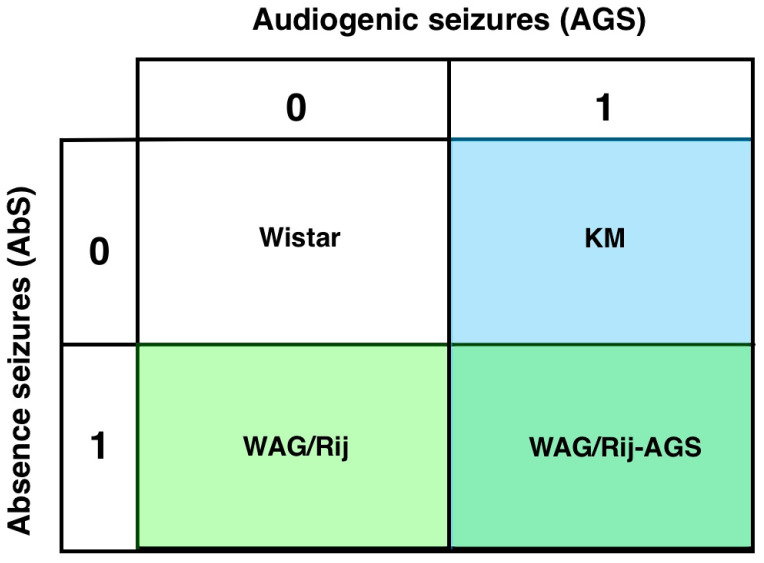
The experimental design: four groups are included in the comparison. WS controls (0-0), KM (only audiogenic seizures, 0-1), WAG/Rij (only absence seizures, 1-0) and rats with mixed types of generalized seizures (WAG/Rij-AGS, 1-1).

**Figure 2 diagnostics-13-00587-f002:**
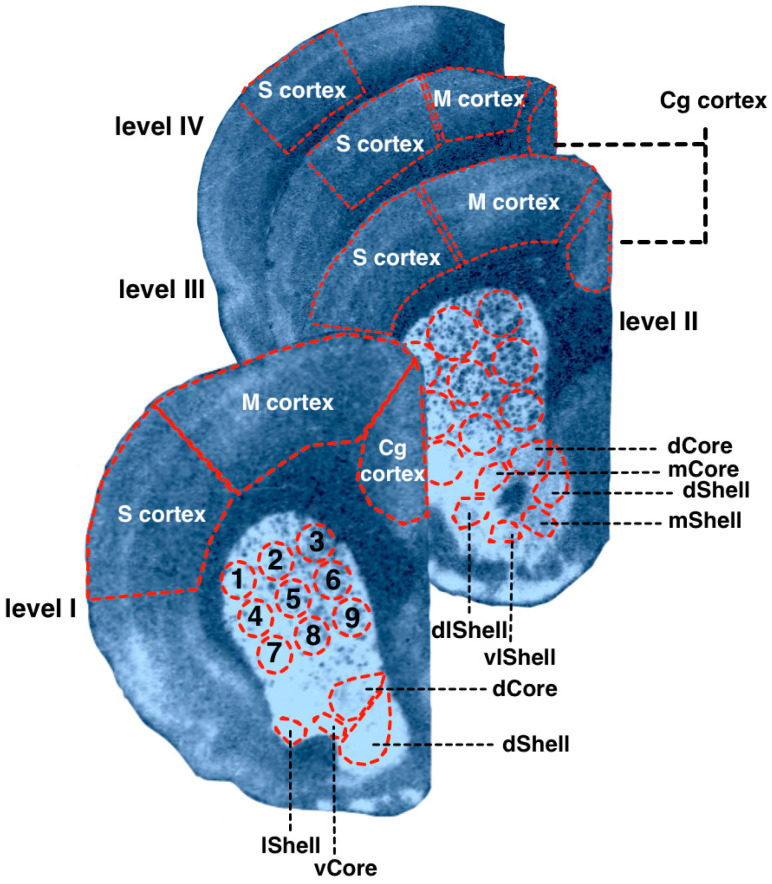
Examples of the autoradiographic imagiing of [3H]spiperone binding densities in the coronal sections of the WS rat brain at the following anatomical levels: +2.4 (level I); +1.68 (level II); −1.08 (level III); −2.16 (level IV) relative to bregma. The rat brain atlas [25] was used to identify and mark brain structures. 1–9—dorsal striatum spatial subregions; Core—nucleus accumbens core; Shell—nucleus accumbens shell; d—dorsal; dl—dorsolateral; l—lateral; m—medial; v—ventral; vl—ventrolateral; Cg—cingulate cortex; M—motor cortex; S—somatosensory cortex.

**Figure 3 diagnostics-13-00587-f003:**
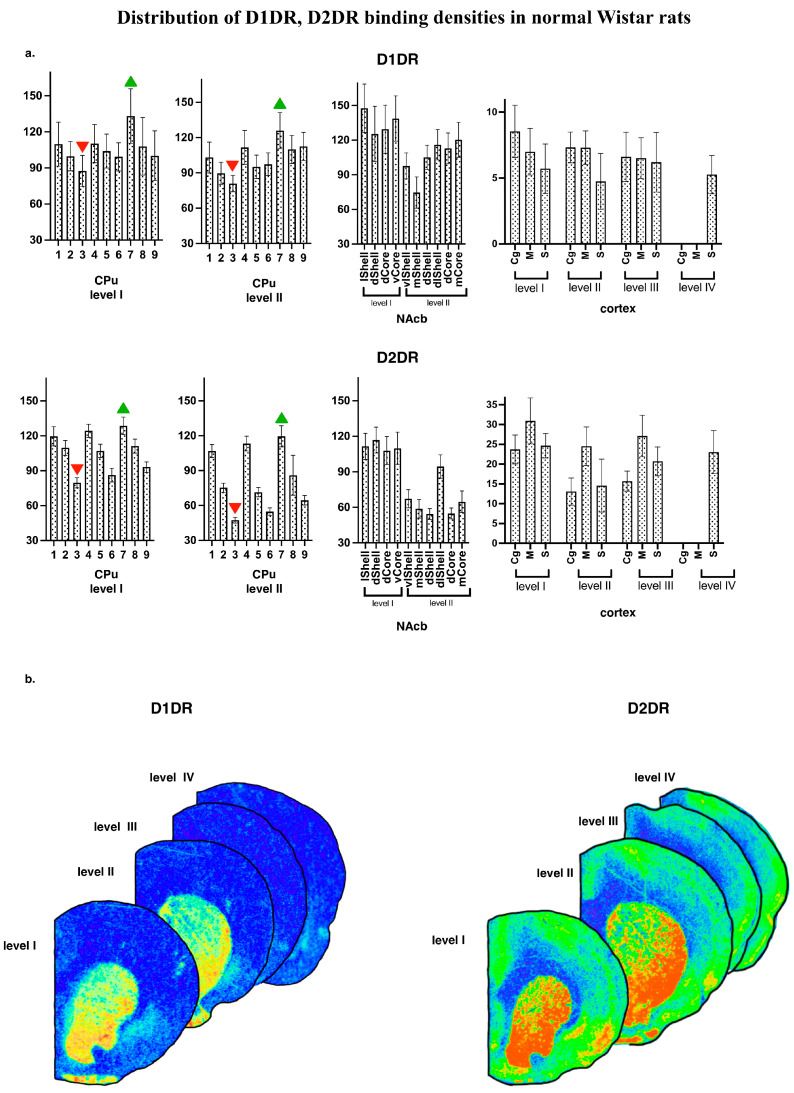
(**a**) Distribution of D1DR, D2DR binding densities in normal WS rats. The values are given as mean ± SEM, in pmol/g of tissue. CPu—caudate putamen; NAcb—nucleus accumbens. For other abbreviations, see Figure 2. (**b**) Examples of the autoradiographic images of [3H]spiperone binding densities in coronal sections of WS rat brain at the following anatomical levels: +2.4; +1.68; −1.08; −2.16 regarding bregma for D2DR and +2.28; +1.8; −0.96; −1.92 for D1DR.

**Figure 4 diagnostics-13-00587-f004:**
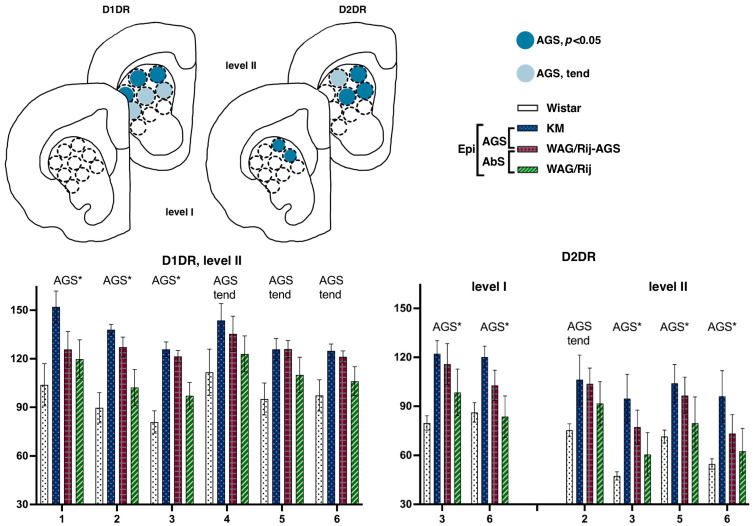
Comparisons of the dorsal striatum binding densities to D1DRand D2DR. The values are given as mean ± SEM, in pmol/g of tissue. For anatomical levels and abbreviations, see Figure 3. Differences are denoted by ‘*’ for *p* ≤ 0.05 and ‘tend’ sign for tendency (0.05 < *p* < 0.10). Effects of audiogenic epilepsy are marked by the “AGS” sign.

**Figure 5 diagnostics-13-00587-f005:**
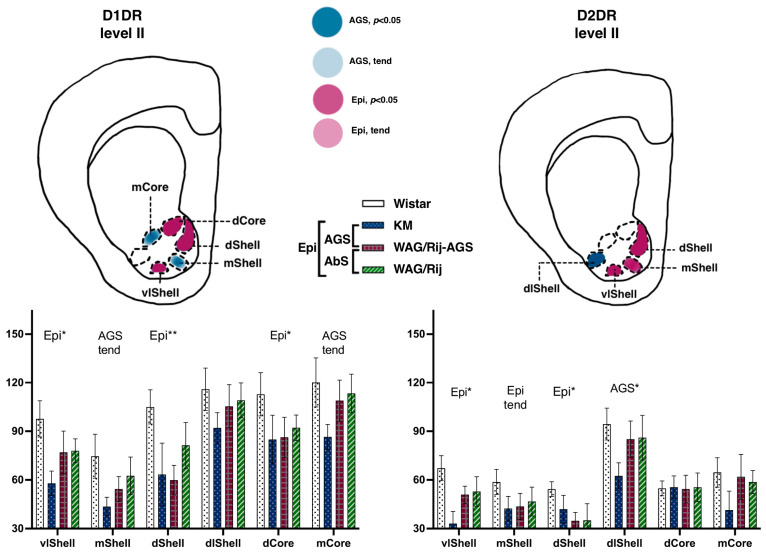
Comparisons of the ventral striatum binding densities to D1DR and D2DR. The values are given as mean ± SEM, in pmol/g of tissue. For anatomical levels and abbreviations, see Figure 3. Differences are shown by stars: * for *p* ≤ 0.05, ** for *p* < 0.01, and ‘tend’ sign for tendency (0.05 < *p* < 0.10). The general effects of epilepsies (a difference between the non-epileptic group and the joint group of all epileptic rats) are marked by ‘Epi’ letters. Effects of audiogenic epilepsy are marked by the ‘AGS’ sign.

**Figure 6 diagnostics-13-00587-f006:**
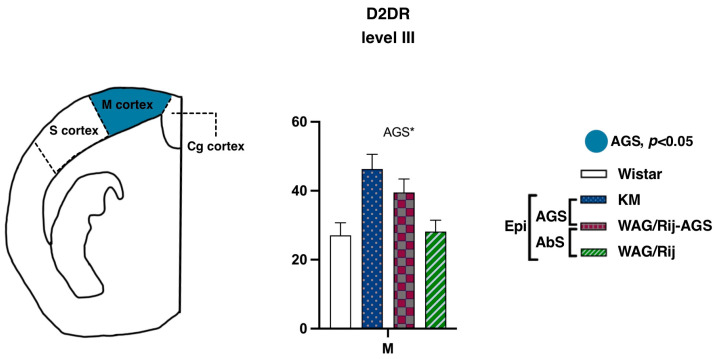
Comparisons of the cortex binding densities to D1DR and D2DR. The values are given as mean ± SEM, in pmol/g of tissue. For anatomical levels and abbreviations, see Figure 3. Differences are shown by ‘*’ for *p* ≤ 0.05. Effects of audiogenic epilepsy are marked by the “AGS” sign.

**Figure 7 diagnostics-13-00587-f007:**
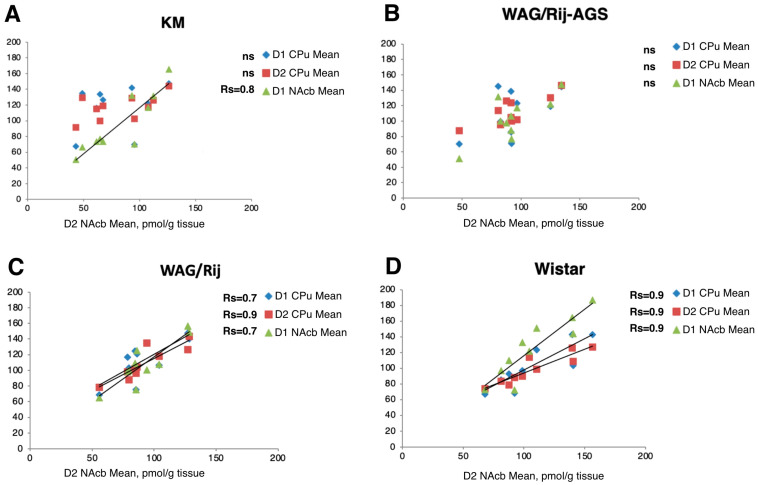
Brain regional binding densities to D1DR and D2DR correlate tightly in two non-AGS groups, but these local correlations are broken in the two AGS-prone cohorts. Axes are mean binding densities, in pmol/g. Each chart represents plots for the rat groups studied (Wistar, WAG/Rij, WAG/Rij-AGS, KM). Each data point represents mean regional D2DR NAcb density binding densities (pmol/g tissue) vs. D1DR or D2DR density in dorsal (blue diamonds for CPU, D1DR; red squares for CPU, D2DR) or ventral striatum parts (green triangles for NAcb, D1DR). In spearman rank order, Rs are given on the charts for significant local correlations (*p* < 0.05), “ns” stands for non-significant ones.

## Data Availability

Datasets are available upon a reasonable request.

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
