# Peer review of "Striatal Patchwork of D1-like and D2-like Receptors Binding Densities in Rats with Genetic Audiogenic and Absence Epilepsies"

_diagnostics, 2023, doi:10.3390/diagnostics13040587_

Round 1

Reviewer 1 Report

At the manuscript “Striatal patchwork of D1-like and D2-like receptors binding 2 densities in rats with genetic audiogenic and absence epilepsies” by Drs. Evgeniya Tsyba   et al, authors studied the density of binding to dopamine receptors D1- and D2- in the brain regions of animals with genetic generalized audiogenic (AGS) and/or absence (AbS) epilepsy (KM, WAG/Rij-AGS, WAG/Rij rats).  Increased D1DR binding density has been found in a number of dorsal striatal subregions of rats with audiogenic epilepsy. Increased D2DR binding density was found in the central and 17 dorsal striatal territories. On the contrary, in the nucleus accumbens, a decrease in the binding density of D1DR and D2DR was found in epileptic animals.

A serious and productive study has been carried out; the data obtained are beyond doubt. I have only minor criticisms:

On Fig. 5 Some letters and numbers are printed on the background of the contours. They are difficult to read. I would suggest moving them, and perhaps changing the font.

By mentioning "Motor Cortex (M)" in the manuscript, does it mean only the primary motor cortex? In this case, it is designated M1. If both primary (M1) and secondary motor cortex (M2) are meant, this should be indicated in the text.

I will be happy to recommend the manuscript for publication after making the corrections indicated above. 

Author Response

Thank You for Your review and encouraging comments on our paper. 

1. On Fig. 5 Some letters and numbers are printed on the background of the contours. They are difficult to read. I would suggest moving them, and perhaps changing the font.

Responce: We have adjusted the lettering on the Figure as suggested.

2. By mentioning "Motor Cortex (M)" in the manuscript, does it mean only the primary motor cortex? In this case, it is designated M1. If both primary (M1) and secondary motor cortex (M2) are meant, this should be indicated in the text.

Responce: «Motor Cortex (M)» here means both primary and secondary motor cortex, pooled during the measurement of optical densities. It is directly indicated now in the section 2.4. Measurements.   

Reviewer 2 Report

In Tsyba et al., the authors provide evidence of striatial patchwork of dopamine D1 and D2-like receptor binding densities in rats with genetic audiogenic and absence epilepsies. Given the role of the dopaminergic system in the brain anti-epileptic system, this study conducted by the authors is of some importance in the field of epileptic behavioral comorbidities. The study is well organized, methods are clearly pointed out, and statistics used are good. However, the authors should cite more recent references to support their findings.

Author Response

The authors are thankful for the reviewing and commenting. The references required have been added, and corresponded text changes marked in color.  

1. However, the authors should cite more recent references to support their findings.
Response: We added some new references (from years 2019-2022) to the Introduction (page1,2) and Discussion (Section 3.2.2.1; page 10).